# Small molecule inhibitors of 15-PGDH exploit a physiologic induced-fit closing system

Wei Huang [1], Hongyun Li[2], Janna Kiselar[3,4], Stephen P. Fink[2], Sagar Regmi [1], Alexander Day[1], Yiyuan Yuan[2], Mark Chance[3,4], Joseph M. Ready [5] ✉, Sanford D. Markowitz [2,6,7] ✉ & Derek J. Taylor [1,6,8] ✉

15-prostaglandin dehydrogenase (15-PGDH) is a negative regulator of tissue stem cells that acts via enzymatic activity of oxidizing and degrading PGE2, and related eicosanoids, that support stem cells during tissue repair. Indeed, inhibiting 15-PGDH markedly accelerates tissue repair in multiple organs. Here we have used cryo-electron microscopy to solve the solution structure of native 15-PGDH and of 15-PGDH individually complexed with two distinct chemical inhibitors. These structures identify key 15-PGDH residues that mediate binding to both classes of inhibitors. Moreover, we identify a dynamic 15-PGDH lid domain that closes around the inhibitors, and that is likely fundamental to the physiologic 15-PGDH enzymatic mechanism. We furthermore identify two key residues, F185 and Y217, that act as hinges to regulate lid closing, and which both inhibitors exploit to capture the lid in the closed conformation, thus explaining their sub-nanomolar binding affinities. These findings provide the basis for further development of 15-PGDH targeted drugs as therapeutics for regenerative medicine.

The enzyme 15-prostaglandin dehydrogenase (15-PGDH) is a key negative regulator of tissue stem cell activity in proliferation and in the repair of tissue injury, whose negative regulation of tissue regeneration is conserved across multiple organs[1]. 15-PGDH acts via its NAD+-dependent enzymatic activity of oxidizing and degrading prostaglandin E2 (PGE2), and related eicosanoids, which provide key trophic support for stem cells in multiple tissues. Specifically, 15-PGDH catalyzes the rate-limiting step to initiate PGE2 degradation by oxidizing the 15-hydroxyl moiety of PGE2 to a 15-keto group, thereby inactivating PGE2 from being able to bind to its receptors[2]. Inhibiting 15-PGDH, either by gene knockout or using small molecule inhibitors, and thereby increasing tissue PGE2 levels, has proved highly effective in therapeutically accelerating tissue regeneration and repair in multiple murine models of organ diseases[1,3-6]. These therapeutic effects range from accelerating mucosal healing in colitis, to markedly accelerating hematopoietic recovery after bone marrow transplantation, to enabling lung recovery from pulmonary fibrosis, and rejuvenating aged muscle mass and strength[1,3-6]. The encouraging success of small molecule 15-PGDH inhibitors in these in vivo disease models reflects their sub-nanomolar affinity for 15-PGDH[1,7]. The inhibitor's high affinity for 15-PGDH is furthermore highly selective, with no detectable effects on other related short-chain dehydrogenases[1]. However, sequence comparisons of the different dehydrogenases do not provide a direct explanation for the high affinity and specificity of 15-PGDH inhibitors. Furthermore, no structure has been solved of 15-PGDH complexed to either an inhibitor or to a natural substrate.

To enable structure-guided drug development of these promising agents, and to advance a fundamental understanding of 15-PGDH

[1]Department of Pharmacology, Case Western Reserve University, Cleveland, OH 44106, USA. [2]Department of Medicine, Case Western Reserve University, Cleveland, OH 44106, USA. [3]Department of Nutrition, Case Western Reserve University, Cleveland, OH 44106, USA. [4]Center for Proteomics and Bioinformatics, Case Western Reserve University, Cleveland, OH 44106, USA. [5]Department of Biochemistry, University of Texas Southwestern Medical Center, Dallas, TX 75390, USA. [6]Case Comprehensive Cancer Center, Case Western Reserve University, Cleveland, OH 44106, USA. [7]University Hospitals Seidman Cancer Center, Cleveland, OH 44106, USA. [8]Department of Biochemistry, Case Western Reserve University, Cleveland, OH 44106, USA. ✉e-mail: joseph.ready@utsouthwestern.edu; sxm10@case.edu; derek.taylor@case.edu

enzymatic function, we have used cryo-electron microscopy (cryo-EM) to solve the structure of human 15-PGDH in the native solution state and when bound independently to two structurally distinct inhibitors. Our results delineate details of the 15-PGDH drug binding pocket and explain the structural basis for the inhibitors' sub-nanomolar inhibitory constants. Inhibitors engage two conserved catalytic residues through hydrogen bonding interactions (S138 and Y151) while simultaneously filling a hydrophobic pocket that may recognize the C16-C20 tail of PGE2. Critically, we also identify structural interactions between the inhibitors and human 15-PGDH that promote an induced closure of a triple-helix lid that fully encapsulates and captures the inhibitor in a buried binding pocket of the enzyme. The lid is stabilized in its closed position through interactions that both inhibitors make with conserved aromatic residues, F185 and Y217 (as numbered in the human enzyme) of 15-PGDH, which also form the hinges of the lid motif. The essential role of these key structural features is validated by their independent incorporation into the mechanism of interaction of two different inhibitors representing two distinct chemical scaffolds. Moreover, changes in 15-PGDH enzyme activity upon substitution of F185 and/or Y217 support that the substrate-induced lid closing mechanism is equally important for binding of the physiologic substrate PGE2 to 15-PGDH. Cumulatively, our results identify a common structural contrivance by which different inhibitors lock the 15-PGDH lid in a closed position, and furthermore reveal a putative mechanism for mediating 15-PGDH enzymatic function.

## Results

### Structural characterization of human 15-PGDH in complex with inhibitor (+)-SW209415

To gain insight into the molecular details of 15-PGDH inhibition, we used cryo-EM and single-particle analysis to solve the structure of the human 15-PGDH protein bound to (+)-SW209415 (Fig. 1a), a second-generation 15-PGDH inhibitor, plus the NADH cofactor. (+)-SW209415 maintains full potency in inhibiting 15-PGDH activity in vivo and in markedly accelerating hematopoietic recovery following bone marrow transplantation, while its 10,000-fold enhancement in solubility affords a major benefit over first-generation inhibitors such as SW033291[4]. The (+)-SW209415-bound human 15-PGDH complex was co-purified prior to cryo-EM grid preparation, using size-exclusive chromatography (SEC). Unique spectral properties recorded for (+)-SW209415 (310 nm), 15-PGDH protein (280 nm), and NADH (350 nm) in the SEC chromatogram were used to confirm complex assembly (Supplementary Fig. 1a).

Despite the relatively small size of the monomeric protein (29 kDa), image and post-processing yielded a cryo-EM map for the 15-PGDH-SW209415-NADH complex at a nominal resolution of 2.4 Å (Fig. 1b and Supplementary Fig. 1b–f). The quality of the cryo-EM map allowed for accurate modeling of protein side-chains, which is correlated with a median Q-score[8] of 0.80 for the comprehensive model (Supplementary Fig. 2). 15-PGDH assembled as a homodimer, which is the proposed physiological form of the enzyme[9]. The dimeric interface consists of an extended helix (α9) and a loop connecting it with the preceding helix (α8) of the polypeptide (Supplementary Fig. 3). The antiparallel arrangement of the two opposing α9 helices from each protomer is maintained by several hydrophobic interactions that stabilize the dimeric arrangement of the enzyme. Specifically, these interactions include F161 with L150′ and A153′ plus A146 with L167′ and A168′. Additional stability is provided by interactions between Y206 of one protomer and L171′ and M172′ of the opposite protomer. These interactions maintain a rigid positioning of the individual α9 helix within each protomer where it butts up against the β5 strand of the classic Rossmann-fold that forms the core of the protein where NADH is bound. The homodimeric interaction is, therefore, likely essential for the proper positioning and stable geometry of three key residues, S138

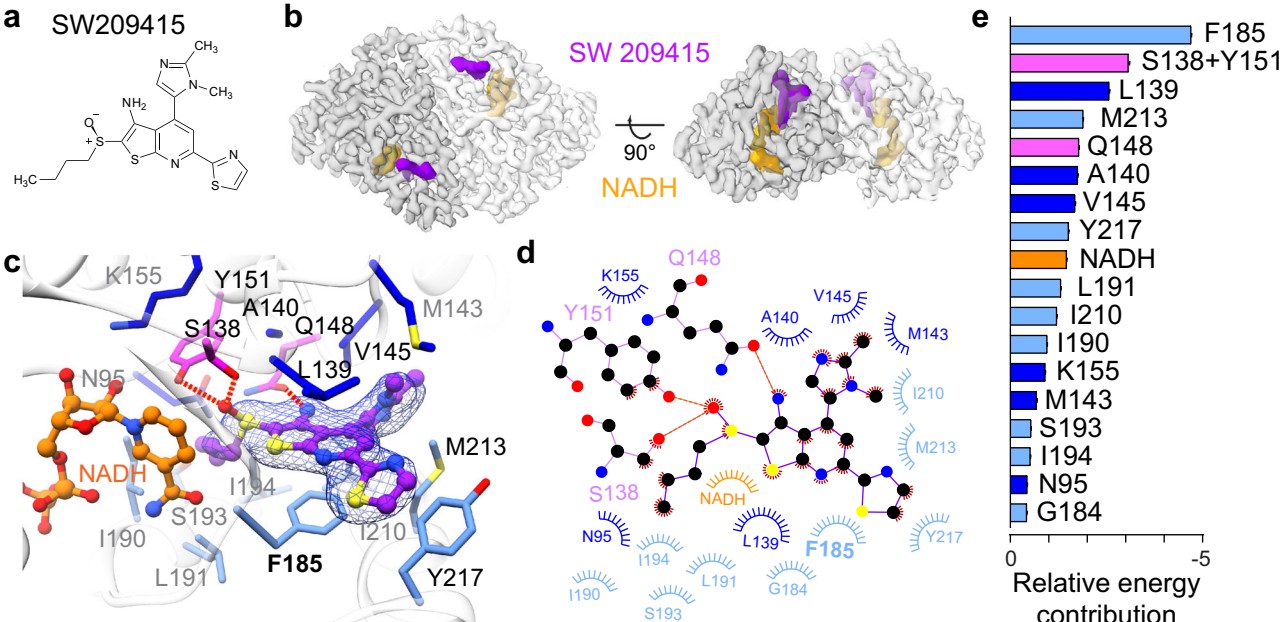

**Fig. 1 | Cryo-EM structure of human 15-PGDH in complex with NADH and (+)-SW209415. a** The chemical structure of (+)-SW209415. **b** The cryo-EM electron density map of the 15-PGDH·NADH·SW209415 complex, with each protomer in the homodimeric assembly differentially colored. (+)-SW209415 and NADH are shown in purple and orange, respectively. **c** Zoom-in view of the drug binding pocket surrounding (+)-SW209415 (purple). Residues forming the drug-binding pocket are color-coded in three groups those in the catalytic core (magenta), those around the catalytic core (blue), and those forming a lid to the pocket (light blue). Hydrogen bonds are shown as red dashed lines. **d** 2D schematic diagram showing contacts between (+)-SW209415 and 15-PGDH. The color code for residues is the same as in **c**. **e** Decomposition of relative energetic contributions of individual 15-PGDH residues to (+)-SW209415 binding as calculated by MM-PBSA. Relative binding energies are presented as mean values ±SEM calculated from 15,000 evenly distributed snapshots from MD simulations.

and Y151, which are required for catalysis, and K155, which is required for cofactor binding as well as reducing the p$K_a$ of Y151, and hence for coordinating interactions between bound substrate and the NAD$^+$/NADH cofactor[10].

Following asymmetric refinement, the cryo-EM structure exhibits strong density for (+)-SW209415 in a well-defined binding pocket in each of the 15-PGDH protomers (Fig. 1b–d and Supplementary Fig. 2d). Strikingly, (+)-SW209415 is buried within the 15-PGDH protein with molecular contacts formed on all sides of the inhibitor. Importantly, (+)-SW209415 prominently binds to the active site of 15-PGDH, where the sulfoxide moiety of the molecule extends into the 15-PGDH catalytic center composed of S138, Y151, and NADH[10]. This interaction is stabilized by hydrogen bonding of the (+)-SW209415 sulfoxide to both S138 and Y151 (Fig. 1c, d). In this position, it is likely that the sulfoxide moiety of (+)-SW209415 mimics the transition state involved in the enzymatic conversion of the cognate PGE2 substrate from an analogous C15-hydroxyl to a C15-keto group[3]. The cryo-EM density further explains why 15-PGDH binding of the (+)-SW209415 enantiomer is significantly tighter than the (-) enantiomer[3]. When docked into the binding pocket of 15-PGDH, the tetrahedral geometry of the sulfoxide in the (-)-SW209415 configuration prevents critical interactions from being formed with S138 and Y151 (Supplementary Fig. 4). This interpretation is consistent with structure-activity relationship studies indicating that all inhibitory activity resides in a single enantiomer of sulfoxide and that removal of the sulfoxide is not tolerated[3].

Cumulatively, the interaction of the sulfoxide of the inhibitor with S138 and Y151 of 15-PGDH contributes significant binding energies and so serves to coordinate (+)-SW209415 interactions with 15-PGDH (Fig. 1e). The thienopyridine moiety of (+)-SW209415 is sandwiched between F185 and L139 (Fig. 1c, d), where π-stacking with F185 provides the largest contribution of all 15-PGDH residues in binding energy (Fig. 1e). The thienopyridine reaches into the catalytic pocket where an amine group at position 3 forms a hydrogen bond with Q148 of 15-PGDH (Fig. 1c–e). Additionally, the (+)-SW209415 thiazole contributes to binding through π-stacking interactions with Y217 and hydrophobic interactions with L139 of 15-PGDH. The imidazole moiety of (+)-SW209415 mediates contacts with other peripheral residues that line the catalytic core of 15-PGDH. Finally, while not as well resolved in the cryo-EM density, the butyl chain of (+)-SW209415 occupies a hydrophobic pocket that is surrounded by I190, L191, and I194 of 15-PGDH. Based on similar physicochemical properties, this hydrophobic pocket is likely important for recognizing and coordinating the n-pentyl group of PGE2. Altogether, our structural description of the inhibitor binding deep within protein provides a vivid explanation for the tight sub-nanomolar binding affinity[1]. However, these findings also pose questions regarding how the compound gains access to the buried drug binding site within 15-PGDH and the generality of this binding mode.

## Identifying a dynamic 15-PGDH lid that closes around bound drug

To interrogate the effects of (+)-SW209415 binding to 15-PGDH, we performed long-time scale (1.5 μs) molecular dynamics (MD) simulations. These data indicate that (+)-SW209415 and residues in the drug binding pocket of 15-PGDH are exceptionally stable in both protomers over the course of the simulation (Supplementary Fig. 5). Molecular Mechanics Poisson-Boltzmann Surface Area (MM-PBSA) method was used to define the relative contributions of individual residues that coordinate (+)-SW209415 binding (Fig. 1e). This analysis highlights two separate regions that reside on opposite sides of the drug binding pocket. One region is composed of the catalytic center and surrounding residues (magenta and blue residues in Fig. 1c, d), and the other is created by the α10-α12 triple-helix motif that spans protein residues F185 to Y217 (light blue residues in Fig. 1c, d). Consistent with our structural data, the triple-helix motif forms a lid-like structure

around the inhibitor where it supports a hydrophobic surface against the drug-binding pocket and a hydrophilic surface on the solvent side. Interactions between the inhibitor and the triple-helix lid highlight the distinct contributions of π-stacking interactions formed between the thienopyridine and thiazole moieties of (+)-SW209415 with 15-PGDH residues F185 and Y217, respectively (Fig. 1e). This triple-helix lid is predicted to regulate access to the 15-PGDH active site (Supplementary Fig. 3).

To further define the predicted functional role of the 15-PGDH drug binding lid, we employed cryo-EM to solve the structure of the 15-PGDH-NADH complex in the absence of inhibitor. With a similar imaging processing workflow (see Methods), we were able to resolve the apo-form to 3.3 Å (Supplementary Fig. 6). The cryo-EM map of the unliganded 15-PGDH shows clear differences from the (+)-SW209415-bound complex described above. In the absence of inhibitor, 15-PGDH forms a looser complex that lacks certain structural features because of enhanced dynamic flexibility. These features are apparent in the 2D class averages and localize mainly to the lid region of the inhibitor binding pocket (Fig. 2a, b). Most notably, in the absence of inhibitor the α10-α12 triple-helix that forms the ligand binding container lid is disordered, and is thus no longer visualized in the cryo-EM analysis; with the lid becoming ordered, and visualized, only upon binding of (+)-SW209415 (comparison of + and – drug panels in Fig. 2a, b).

To delineate the relationship between inhibitor binding and functional dynamics of 15-PGDH, we performed long-time scale molecular dynamic (MD) simulations of the enzyme following the extraction of inhibitor from the model derived from our cryo-EM structure. Superpositioning of snapshots from the 4.5-μs MD simulations reveals that, while most of the protein remains static during this extended time scale (<2 Å deviation), the α10–12 triple-helix lid of the drug binding container deviates significantly from the stable structure that is observed in the inhibitor-bound state and demonstrates the largest fluctuation during the simulation (Fig. 2c, d). Notably, conformational changes of the lid include deviations of secondary structure features. Thus, the MD simulations recapitulate the cryo-EM findings that the lid becomes disordered in the absence of inhibitor binding. Together these results show that binding of (+)-SW209415 stabilizes the lid structure of the α10-α12 triple-helix to create a stable configuration of (+)-SW209415 complexed with 15-PGDH. The key elements of this stabilization are interactions of (+)-SW209415 with F185 and Y217, interactions that serve to both regulate the closing of the lid and to maintain it in the closed position. This structural observation explains findings from our group that binding of (+)-SW209415 to 15-PGDH is essentially irreversible, and that the ligand cannot be dialyzed off, even over 24 h (data not shown). Another intriguing observation that emerges from these long MD simulations is that the container lids of the two protomers exhibit differences in their degree of fluctuations in the homodimer. For example, the lid of one protomer adopts a closed position relatively early, while the other one continues to fluctuate throughout the extended simulation (Fig. 2d). A static structure taken from the trajectory (~600 ns) illustrates this dynamic difference can be attributed to contacts formed between part of the lid of one protomer and the dimeric interface of the opposite protomer (Supplementary Fig. 5c). Residues within the lid (Y203 and Y206) form interactions with the opposite protomer (Y116, A168, L171, and M172) to help stabilize the lid in a closed position. Such an asymmetric configuration could result in at least one protomer always being available for drug or substrate binding while maintaining overall structure in an energetically favorable state.

Comparing our 15-PGDH-SW209415-NADH structure to that of 15-PGDH bound only to NAD$^+$, as previously solved by X-ray crystallography[7], identifies a shift of NADH toward (+)-SW209415 (Supplementary Fig. 7a). Despite this shift in cofactor displacement, there is less than a 2 Å root mean square displacement (RMSD) of the

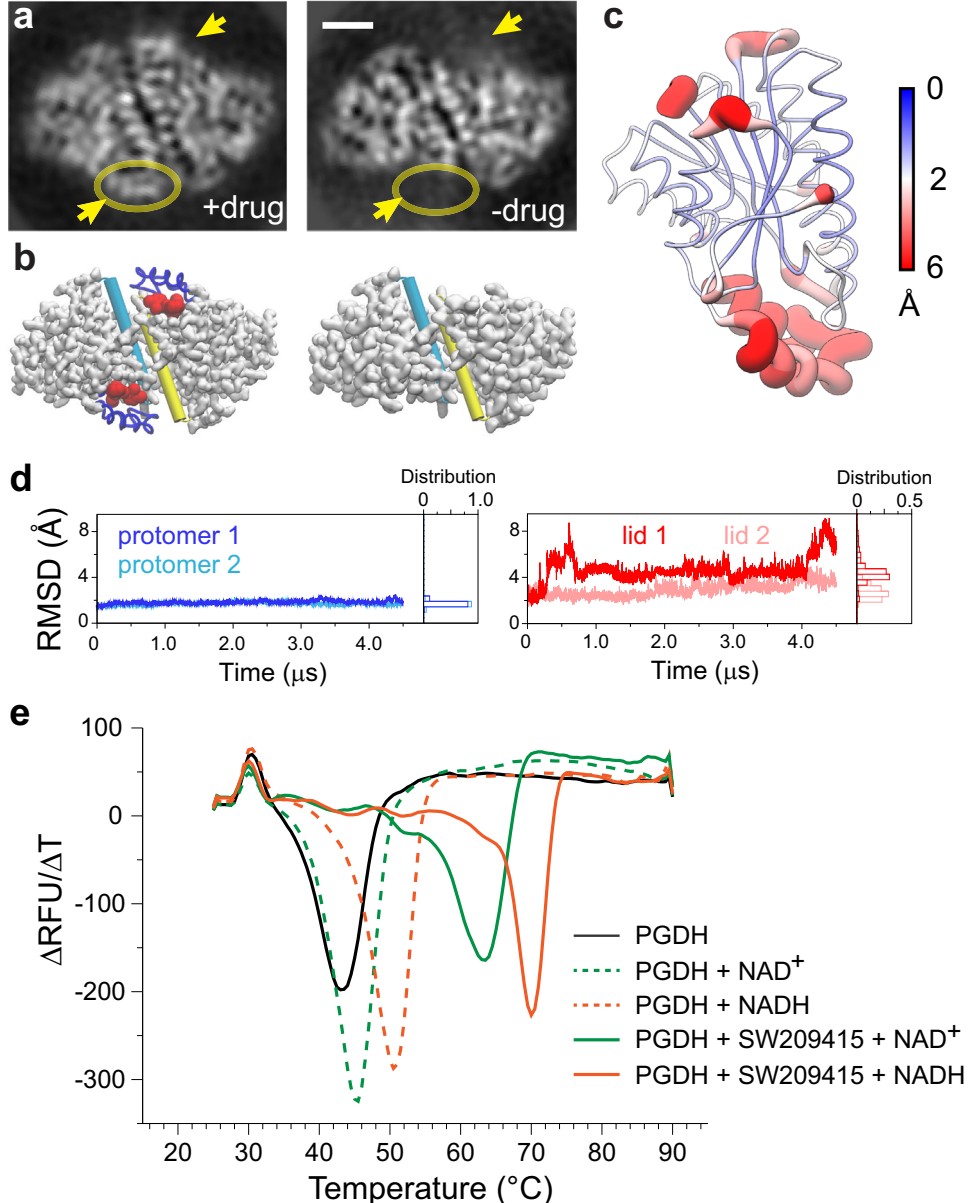

**Fig. 2 | The 15-PGDH drug-binding lid is dynamic and disordered in the absence of inhibitor.** **a** A comparison between the 2D class averages of human 15-PGDH with (left) and without (right) (+)-SW209415. The presence or absence of density to account for the inhibitor and the drug binding lid are highlighted with yellow arrows. Scale bar is 20 Å. **b** Structural illustration of the conformational changes in the absence of inhibitor (shown as red sphere). The obvious conformational change is the loss of the drug binding clamp (shown as blue ribbon). The α9 helices involved in dimer interface are shown as cylinder tube in cyan and yellow for individual subunits respectively. **c** Root mean square fluctuation (RMSF) of Cα atoms from a 4.5-μs trajectory MD simulation of 15-PGDH without inhibitor. RMSF values are displayed on the ribbon representation of a 15-PGDH protomer. The radius of the ribbon is proportional to RMSF values of individual residues. **d** Time evolution of RMSD of drug binding lid shows large deviations from the starting structure (right), while the rest of the protein does not deviate significantly from the starting structure (left). Analysis of individual protomers within the homodimer is overlaid and labeled accordingly. **e** Thermal melting profile of 15-PGDH alone (black), 15-PGDH with NAD+ (green dashed line) or NADH (orange dashed line), and 15-PGDH in the presence of (+)-SW209415 with NAD+ (green solid line) or NADH (orange solid line).

overall drug binding pocket geometry of 15-PGDH when comparing the two structures. These RMSD differences within the protein are, primarily, due to changes in side-chain rotamers. This result suggests that cofactor positioning is dynamic, and that its precise positioning differs depending upon occupancy of different ligands in the active site and/or the distinct electrostatic potential of NAD+ versus NADH (Supplementary Fig. 7b). Indeed, our data indicate that the nature of the cofactor is, of itself, an important contributor of complex stability, since 15-PGDH demonstrates a higher melting temperature in the presence of NADH ($T_m$ = 51.6 °C) as compared to NAD+ ($T_m$ = 45.8 °C) (Fig. 2e). Addition of inhibitor induces an even greater shift in melting

temperature of 15-PGDH, but this again is even more pronounced in the presence of NADH ($T_m$ = 70.0 °C) over NAD+ ($T_m$ = 63.5 °C) (Fig. 2e). Based on our structural observations, the increased melting temperature of 15-PGDH when bound to inhibitor likely reflects the enhanced stability of the enzyme when the triple-helix lid is closed around (+)-SW209415. In contrast, the difference in stability for NADH vs NAD+ binding is most likely associated with differences in physico-chemical properties, including non-covalent van der Waals and hydrogen bonding interactions, related to cofactor identity and subtle differences in the precise location of each cofactor near the 15-PGDH catalytic site.

# a

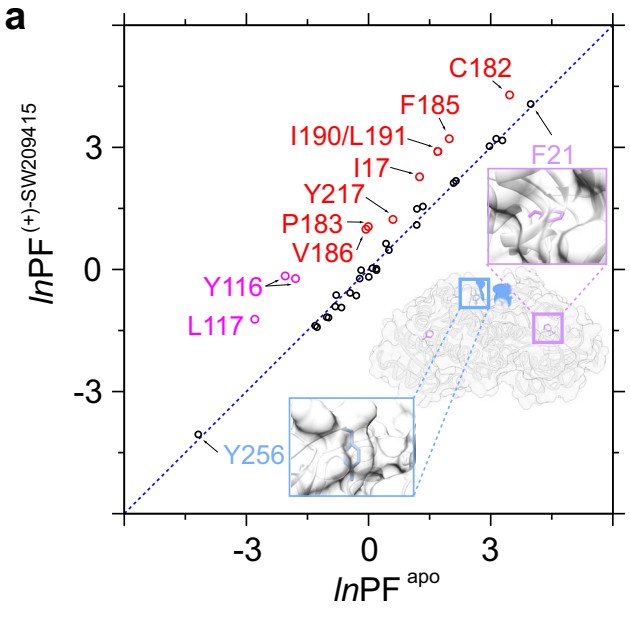

# b

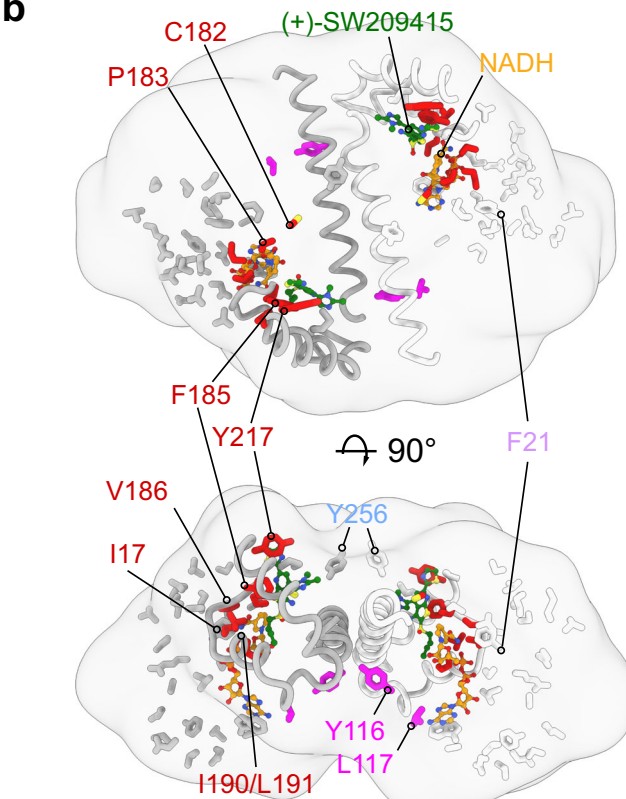

**Fig. 3 | Hydroxyl radical footprinting of human 15-PGDH in the absence and presence of (+)-SW209415. a** Scatter plot for Protection Factor (PF) analysis of the hydroxyl radical footprinting data with (y axis) and without (x axis) (+)-SW209415 bound. The relative PF value of a residue indicates its location within the protein, as illustrated by two examples. F21 is buried deep inside the protein, while Y256 is on the surface of the protein. Residues on the upper left triangle become more protected in the presence of (+)-SW209415. Residues involved in either NADH or drug binding are colored in red, residues involved in the dimeric interface are colored in magenta. **b** Mapping of the probes identified in hydroxyl radical footprinting on the (+)-SW209415-bound human 15-PGDH structure. These residue probes are shown as stick with the same color code used in panel a. NADH and (+)-SW209415 are depicted in orange and green ball-and-stick models. Residues that are more protected with (+)-SW209415 are clustered around the drug-binding pocket and, specifically, the nicotinamide moiety of NADH. Two additional residues, Y116 and L117, localize to the 15-PGDH homodimeric interface and are more protected when (+)-SW209415 is bound.

## Hydroxyl radical footprinting confirms predictions of the cryo-EM structure

To further investigate the functional dynamics of the enzyme in solution, we conducted hydroxyl radical footprinting (HRF) experiments of NADH-bound 15-PGDH in the presence and absence of (+)-SW209415. Each complex was exposed to synchrotron X-rays and a Protection Factor (PF) was calculated for the rates of hydroxyl radical modification for individual amino acids, which was derived by mass spectrometry[11,12]. Application of this technique presents a comprehensive comparison of the two states being investigated (Fig. 3a and Supplementary Table 1). First, the *ln*PF values individually describe the relative buried extent of corresponding amino acid probes within the

functional protein[11]. For example, F21 is buried deep inside the core of the protein and is, therefore, associated with the highest *ln*PF value; whereas, Y256 resides at the protein surface and shows the lowest *ln*PF value (Fig. 3a). Secondly, the PF analysis collectively reports on structural rearrangements between the two states by identifying residues whose PF changes between the (+)-SW209415-bound versus unbound conformations of 15-PGDH. The HRF results suggest that the majority of the 15-PGDH protein maintains a folded state with most amino acids exhibiting minimal environmental changes upon (+)-SW209415 binding (black circles, Fig. 3a). Notably, a cassette of residues shifts to a more protected state upon (+)-SW209415 binding (colored circles, Fig. 3a). Mapping of these amino acid probes onto the 3D structure indicates that the residues protected by (+)-SW209415 binding fall into two categories that are defined by localization. The first group includes residues that form direct molecular contacts with (+)-SW209415 in the cryo-EM structure (red circles, Fig. 3a; red residues, Fig. 3b). These residues prominently include F185 and Y217, which play a key role in ordering the lid, and I190 and L191 that interact with the butyl side chain of the inhibitor. The second group similarly reports increased protection with binding of (+)-SW209415, but cluster onto the surface of the protein (magenta circles, Fig. 3a; magenta residues, Fig. 3b). Many of these amino acids (e.g., Y116 and L117) reside at the dimeric interface of the enzyme when the lid is closed around (+)-SW209415, Y116 forms a hydrogen bond with Y203 of the opposite protomer (Supplementary Fig. 8). Thus, this second set of protected residues suggests that binding of (+)-SW209415 stabilizes the homodimerization interface. Thus, the complementary results of HRF, performed under ambient conditions, independently support multiple key conclusions of the cryo-EM structural analysis.

## Binding of SW222746 a next-generation 15-PGDH inhibitor with a distinct chemical scaffold

To examine the generality of the molecular interactions that govern 15-PGDH inhibition, we expanded our investigation to include a newer generation of 15-PGDH inhibitors, SW222746[13] that represents an entirely different chemical scaffold (Fig. 4a). SW222746, has a similar IC50 as (+)-SW209415, but is based on a quinoxaline scaffold that is chemically distinct from the thienopyridine-based structure of (+)-SW209415. We used cryo-EM to define the molecular interactions of SW222746-bound to 15-PGDH, which were interpreted from the structure determined with an average resolution of 2.9 Å (Fig. 4b). Strikingly, SW222746 occupies a nearly identical pocket as that of (+)-SW209415, while using different chemical groups to maintain critical interactions with key 15-PGDH residues. Specifically, as was the case with (+)-SW209415, SW222746 is fully encapsulated in the drug binding pocket with the α10-α12 lid of 15-PGDH closed around it. Similar to (+)-SW209415, the closing of the lid again appears to be orchestrated through π-stacking interactions coordinated through

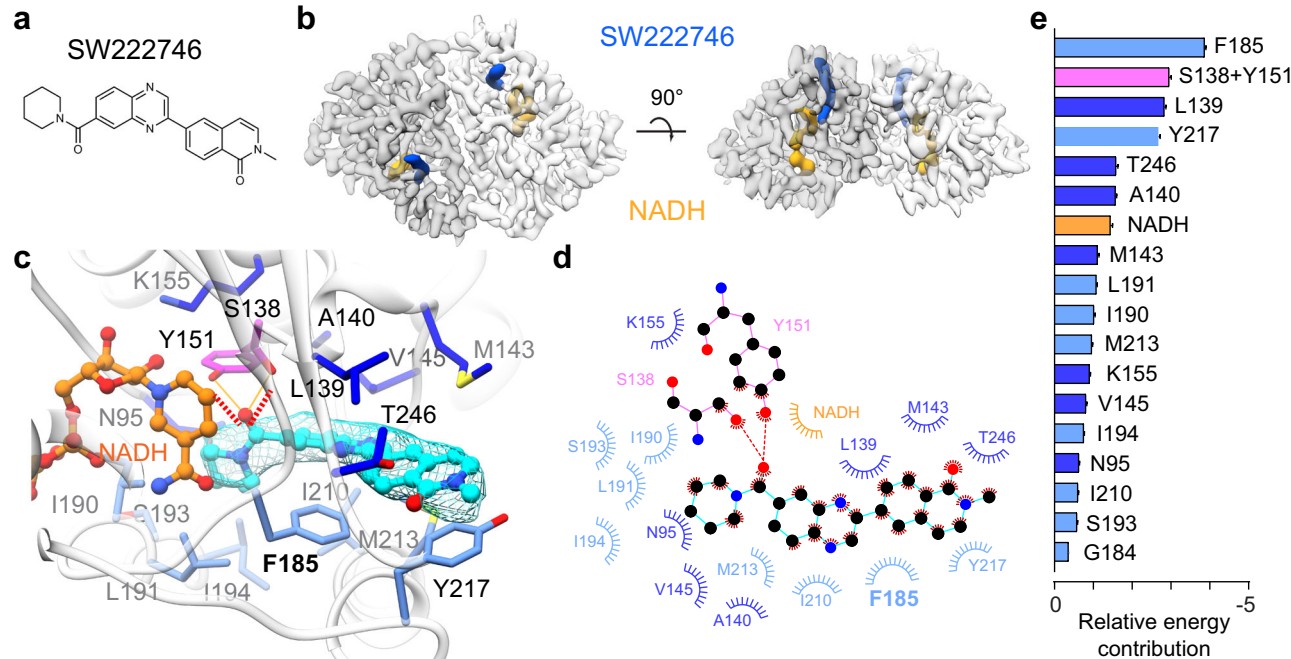

**Fig. 4 | Cryo-EM structure of human 15-PGDH in complex with NADH and SW222746. a** The chemical structure of SW222746. **b** The cryo-EM electron density map of the 15-PGDH·NADH·SW222746 complex, with each protomer in the homodimeric assembly differentially colored. SW222746 and NADH are displayed in light blue and orange respectively. **c** Zoom-in view of the drug binding pocket of SW222746 (cyan). Residues forming the drug-binding pocket are color-coded in three groups those in the catalytic core (magenta), those around the catalytic core

(blue), and those forming a lid to the pocket (light blue). Hydrogen bonds are shown. **d** 2D schematic of contacts between inhibitor SW222746 and 15-PGDH. The color code for residue is the same as **c**. **e** Decomposition of relative energetic contributions of individual 15-PGDH residues to SW222746 binding as calculated by MM-PBSA. Relative binding energies are presented as mean values ±SEM calculated from 15,000 evenly distributed snapshots from MD simulations.

F185 and Y217 of 15-PGDH with the quinoxaline and quinoline-2(1H)-moieties, respectively, of SW222746. The amide carbonyl group of SW222746 replaces the (+)-SW209415 sulfoxide and maintains the specific recognition of the catalytic core of the enzyme through, again, a hydrogen bonding network formed with the critical S138 and Y151 residues (Fig. 4c, d). The piperidine moiety of SW222746 replaces the butyl chain of (+)-SW209415 to maintain hydrophobic interactions as well.

Although (+)-SW209415 and SW222746 occupy similar positions in the drug binding pocket of 15-PGDH, there are several key differences in molecular interactions formed between each inhibitor and the enzyme (Supplementary Fig. 9). While both inhibitors conserve many of the same van der Waals interactions with 15-PGDH residues in the binding pocket, MM-PBSA analysis reveals a rebalancing of contributions from those individual amino acids when binding the different inhibitors (Fig. 4e). For example, the thienopyridine of (+)-SW209415 reaches into the catalytic pocket to form a hydrogen bond (amine group at position 3) with Q148 of 15-PGDH (Fig. 1c–e), whereas SW222746 does not form any interactions with Q148. Additionally, the elongated quinoxaline and quinoline-2(1H)-one moiety of SW222746 form interactions with T246 within the β7 strand of 15-PGDH that is not observed for (+)-SW209415 binding to the protein, and interactions with Y217 are stronger for SW222746 than for (+)-SW209415 (Fig. 1e vs 4e). Another comparison between the two inhibitor-bound structures reveals a displacement in the relative positioning of the two protomers within the assembled homodimer, with larger Cα displacements relative to the axis of symmetry observed in the SW222746-bound complex (Supplementary Fig. 10a). Finally, the placement of the two inhibitors is superimposable within the individual binding pocket of each protomer, the geometry and positioning of the NADH cofactor are altered in the two structures (Supplementary Fig. 10b).

## F185 and Y217 are key functional lid residues for binding both drug and PGE2 substrate

To further dissect the functional role of the two conserved residues, F185 and Y217, we generated 15-PGDH with single (F185A or Y217A) or double alanine substitutions (F185A + Y217A) and analyzed interactions of the mutant proteins with (+)-SW209415 and SW222746. Thermal melting data revealed that individual mutations resulted in slightly reduced protein thermal stability (by 1.0–1.5 °C), while the double-mutant exhibited a slightly more enhanced effect (by 3.5 °C) (Fig. 5a). All mutant proteins were further stabilized with addition of NADH (by 2.0–3.5 °C), indicating that the mutant proteins fold properly. However, the added thermal stability from NADH binding to the mutants was less pronounced than that observed for wild-type protein (5.0–5.5 °C), suggesting potential interaction between the pocket created by the closed lid and the structurally adjacent cofactor binding site. Most importantly, the thermal melting data confirmed the key role that the F185 and Y217 lid residues play in capturing the two inhibitor molecules. In particular, compared to wild type 15-PGDH, the double mutant lost a marked 13.0 °C of thermal stabilization from binding to (+)-SW209415 (21.5 °C in wild type versus 8.5 °C in the double mutant) and lost an even greater 18.0 °C of thermal stabilization from binding to SW222746 (22.5 °C versus 4.5 °C). Effects of the individual F185A mutant were also pronounced, reducing the thermal stabilization of binding to (+)-SW209415 and to SW222746 by 12.5 °C and 10.0 °C, respectively. In contrast, the individual Y217A mutant had no effect on the thermal stabilization of binding to (+)-SW209415, but reduced thermal stabilization of binding to SW222746 by 6.0 °C. These observations illustrate the dominant contribution of F185 to both (+)-SW209415 and SW222746 binding that was highlighted by our MM-PBSA calculations and also support the greater importance of interaction with Y217 that these calculations predicted for binding of SW222746 versus (+)-SW209415 (Figs. 1e, 4e). We note that while F185A

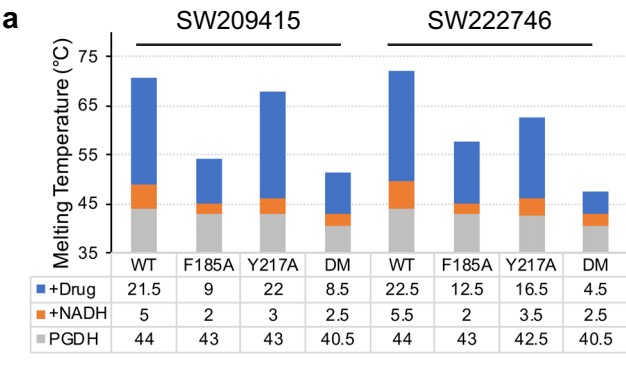

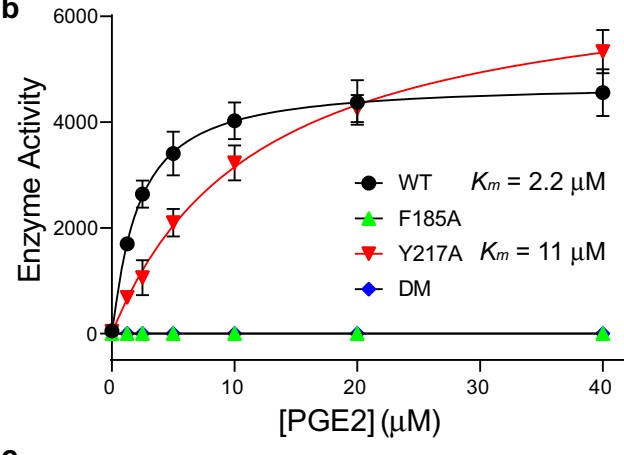

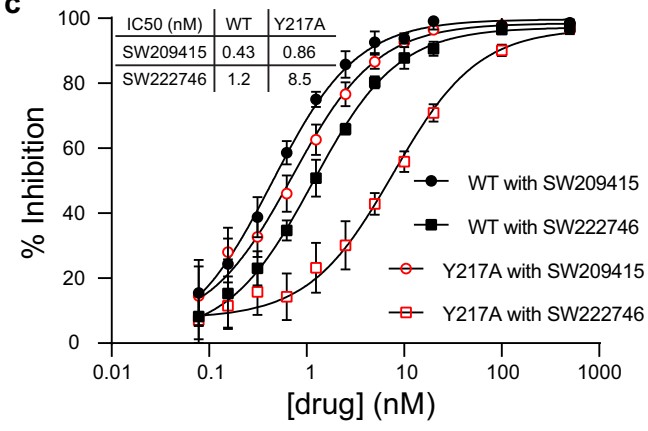

**Fig. 5 | Conserved F185 and Y217 residues contribute both to 15-PGDH enzymatic function and to 15-PGDH binding of inhibitors. a** Shown is the melting temperature ($T_m$) of wild type and mutant 15-PGDH proteins (gray bar), along with increments of $T_m$ acquired upon addition of NADH (orange bar), or upon addition of both NADH plus drug (blue bar). F185A and Y217A are individual point mutants and DM is the F185A + Y217A double mutant. **b** Michaelis–Menten enzyme kinetics for wild type and mutant 15-PGDH proteins in oxidizing PGE2. Wild-type PGDH has a $K_m$ of 2.2 μM and Hill coefficient $n_H = 1.0$; the Y217A mutant has a $K_m$ of 11 μM and $n_H = 1.0$; and F185A and the double mutant are both inactive. Data are presented as mean values ±SD ($n = 3$). **c** Percent inhibition (0–100%) of wild type and F217A mutant 15-PGDH enzyme activity versus concentration for (+)-SW209415 and SW222746, with a concentration in log scale. At a tested 15-PGDH protein concentration of 2 nM, IC50 values for (+)-SW209415 for wild type and Y217A are 0.43 and 0.86 nM, respectively, and IC50 values for SW222746 for wild type and Y217A are 1.2 and 8.5 nM, respectively.

and Y217A mutations each impairs, neither eliminates, binding of (+)-SW209415 and SW222746 inhibitors to 15-PGDH.

We next used enzyme kinetics to interrogate the role of F185 and Y217 in normal 15-PGDH function. The results indicate that the Y217A

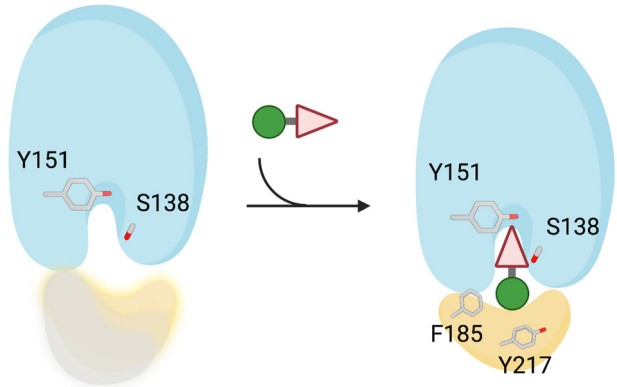

**Fig. 6 | Schematic illustrating the proposed mode of action involving 15-PGDH inhibition.** (+)-SW209415 and SW222746 inhibitors possess different chemical groups that each specifically recognize the catalytic core of 15-PGDH. Binding of inhibitor substituents (triangle) to the catalytic center (S138 and Y151) positions the inhibitor within the enzyme. Binding of either inhibitor then induces a hydrophobic collapse of residues that form the lid (yellow) of the enzyme binding site to fully encapsulate the inhibitors in the protein binding pocket. The conserved F185 and Y217 residues reside at either end of the 15-PGDH lid and form interactions with other inhibitor substituents (green ball) to maintain the lid in the closed position.

mutation reduces 15-PGDH affinity for PGE2, as evidenced by a fivefold increase in $K_m$; whereas, the F185A single and the F185A/Y217A double mutants were completely inactive, likely due to an even greater loss of ability to bind substrate by 15-PGDH (Fig. 5b and Supplementary Fig. 11). These results strongly suggest that PGE2 also interacts with these lid residues and thus directly implicate the lid closure mechanism as integral to the interaction of 15-PGDH with it physiologic PGE2 substrate. Moreover, enzyme kinetics independently confirmed the impaired binding of the Y217A mutant to SW222746, as evidenced by a sevenfold increase in the IC50 for inhibiting the Y217A mutant compared to wild type enzyme (Fig. 5c). With this more sensitive assay, a weaker impairment for Y217A binding to (+)-SW209415 was also detectable, as shown by a twofold increase in IC50 for inhibiting the mutant relative to wild type 15-PGDH.

Altogether, these biochemical data confirm the structural information that lid residues F185 and Y217 are both directly involved in 15-PGDH binding of (+)-SW209415 and SW222746. Moreover, these functional data reveal that F185A and/or Y217A mutations adversely affect 15-PGDH interaction with PGE2, thereby supporting a physiologic role of these residues and of the lid-closing mechanism in the interaction of 15-PGDH with the native PGE2 substrate.

## Discussion

Small molecule 15-PGDH inhibitors represent a promising therapeutic strategy in elevating PGE2 levels to promote tissue stem cell proliferation and to accelerate tissue regeneration and repair, as demonstrated by efficacy in multiple mouse models of human diseases that include colitis, pulmonary fibrosis, aplastic anemia, sarcopenia, and hematopoietic recovery after bone marrow transplant[4–6,14–16]. In this investigation, we report the structures of the 15-PGDH protein bound to small molecule inhibitors to define the molecular interactions that are responsible for binding and inhibition. In addition, we have identified a dynamic region of 15-PGDH that acts as a lid that is conformationally regulated by ligand binding and that is exploited by two unrelated chemical scaffolds to enable potent inhibition of 15-PGDH activity (Fig. 6; Supplementary Movie 1). Molecular interactions of both inhibitors with residues lining the catalytic site of 15-PGDH, specifically S138 and Y151, are likely essential for the initial binding and positioning of the inhibitor. The stabilization and closing of the 15-PGDH lid are events orchestrated by molecular contacts formed between the

inhibitors and, primarily, two key conserved and hydrophobic residues, F185 and Y217, which reside at either end of the triple-helix motif (helices α10-α12) that forms the lid of the drug binding pocket. By essentially locking the 15-PGDH lid in its closed position, the key interactions with F185 and Y217 enable both inhibitors to completely and potently inactivate 15-PGDH enzymatic activity. Moreover, the relative proximity of the two F185 and Y217 hinge residues creates the salient vulnerability that, via similar structural stratagems, both small molecule inhibitors potently contrive to exploit.

In support of this model, our data identify energetically dominant interactions formed between the F185 hinge residue and distinct chemical moieties of each of (+)-SW209415 and SW222746. In a similar fashion, we find that Y217 contributes π-stacking interactions with both inhibitors, particularly with SW222746, at the second hinge position at the opposite end of the 15-PGDH lid. Furthermore, we find that the resulting closed lid engulfs the inhibitors in a deep pocket that is surrounded on all sides by protein motifs. In sum, these structural features can explain the high affinity binding and potency of the inhibitors (IC50 ~1.1 and 3.0 nM for (+)-SW209415 and SW222746, respectively[4,17]).

In the absence of an inhibitor, our data indicate that this lid of 15-PGDH successively swings between open-and-closed states in solution. We predicted that this dynamic oscillation is likely to be equally important in regulating sampling, binding, catalysis, and release of the physiologic substrates and products. The high conservation of F185 and Y217 across species further supports that by regulating lid conformation, these two residues could play a key physiological role in controlling access to the 15-PGDH catalytic site by the native prostaglandin substrates (Supplementary Fig. 12). Our biochemical data now directly implicate the F185 and Y217 lid residues in the physiologic interaction of 15-PGDH with PGE2, an interpretation that is further supported by docking studies and homology models that predict key interactions between both of these residues and bound PGE2 substrate[7,18]. Thus, the potency of the small molecule inhibitors reflects their subverting an essential enzymatic mechanism that evolved to regulate 15-PGDH interaction with the physiologic substrate. One interesting hypothesis is that the dehydrogenation of PGE2 to 15-keto-PGE2 may result in a conformational change that weakens or disrupts the interaction between the 15-keto molecule and F185 and Y217 of 15-PGDH, thereby allowing the lid to reopen and release the 15-keto-PGE2 product. Determining details of the molecular mechanism of normal catalysis and product release will require further testing, which will be assisted by developing structural models of PGE2 substrate and/or product bound to 15-PGDH.

Interestingly, the previously solved 15-PGDH-NAD[+] crystal structure depicts the drug-binding lid as artificially stabilized in its closed state[7]. In that structure, since there is no inhibitor or substrate bound to the enzyme, the closed position of the 15-PGDH lid appears to be an artifact related to crystal packing and/or the dehydration process that is needed for structure determination by X-ray crystallography. This observation likely explains the reported inability to co-crystallize 15-PGDH with a small molecule in vitro, as the forced closure of the lid in the crystal would preclude access and binding of the inhibitor. This artifact was however not a factor in our cryo-EM single-particle analysis of the 15-PGDH protein in solution.

In addition to targeting the 15-PGDH lid, we find that both inhibitors take advantage of other structural features of 15-PGDH. The hydrophobic cavity surrounding I190 and L191, and adjacent to the catalytic site residues of 15-PGDH, provide contacts that enhance binding energies and further stabilize interactions for both inhibitors investigated. Another important feature in the two inhibitor-bound states of 15-PGDH involves interactions with the Q148 residue that resides near the catalytic center of the enzyme. An interaction between Q148 and (+)-SW209415 helps to position the bound inhibitor in the 15-PGDH active site. Homology modeling and docking studies have

been used to propose the involvement of Q148 in the catalytic oxidation of the physiologic PGE2 substrate of 15-PGDH[19]. While Y151 and S138 of 15-PGDH are strictly conserved across the larger family of short-chain dehydrogenase/reductase (SDR) enzymes, the Q148 position is surprisingly variable and is represented by glutamine, glutamic acid, histidine, or asparagine[19]. Site-directed mutagenesis studies indicate that Q148 of 15-PGDH serves as a hydrogen bond acceptor for the oxidation reaction of PGE2[20–22]. For example, substitutions that maintain hydrogen bonding capabilities (Q148E/H/N) have little effect on enzyme activity, whereas the Q148A mutation results in a complete loss of activity[19]. These data define the role of Q148 in optimally positioning the bound PGE2 substrate and/or in stabilizing the transition state intermediate and/or product during the oxidation reaction. Interestingly, our data indicate that the binding of SW222746 depends primarily on Y217, F185, L139, S138, and Y151 of 15-PGDH and with little to no contributions from Q148 (Fig. 4).

In addition to elucidating the structural interactions that enable the binding of 15-PGDH and inhibitors, our studies also identify regions of the 15-PGDH homodimer that are flexible and dynamic. These regions correlate with subtle differences in protein conformation when comparing the two inhibitor-bound structures of 15-PGDH, and these differences are very likely coincident with functional motions needed for PGE2 processing. For example, differences in the relative arrangement of the individual protomers of each structure suggest that alterations at the homodimeric interface are an important feature of enzyme function. Additionally, changes in precise NADH/NAD[+] positioning among structures solved indicate that cofactor placement and rearrangement may contribute to the dehydrogenation reaction of the native substrate. Taken together, it is plausible that the (+)-SW209415- and SW222746-bound structures represent different snapshots of intermediate-like states that are recapitulated in the PGE2 processing pathway. For example, the enzymatic reaction of 15-PGDH involves the removal of a proton from the hydroxyl group of PGE2 and the removal of a hydride from carbon 15 (C15). The transition state of this reaction would, therefore, have a partial negative charge on the oxygen and a partial positive charge on C15. One implication, stemming from the structure of (+)-SW209415 in complex with 15-PGDH, is that the tetrahedral sulfoxide, as positioned in the catalytic site, mimics the PGE2 intermediate state by having a partial positive charge on the sulfur and a partial negative charge on the oxygen. By contrast, the planar geometry of the carbonyl group of SW222746 may more closely resemble the 15-keto-PGE2 bound complex. Long-time scale molecular dynamic (MD) simulations of 15-PGDH suggested that the lids on the opposite protomers might differ in the poses adopted during fluctuations without inhibitor bound. However, our kinetics studies showed the relation of 15-PGDH enzyme activity versus PGE2 concentration fits a simple sigmoidal curve characterized by Hill coefficient of 1 for both the wild-type and Y217A substituted 15-PGDH proteins. It is thus most likely that the two protomers simply act independently in their binding to inhibitor or to substrate.

15-PGDH belongs to an extensive family of SDR proteins[23]. Our analysis of 15-PGDH helps to more globally understand the structure-function relationship of this class of enzymes. For example, despite the low sequence identities, several SDR proteins use a conserved Rossmann-fold core domain to uniformly recognize NAD[+]/NADH cofactor and maintain the catalytic triad of Ser-Tyr-Lys[23]. Moreover, the defined lid mechanism we have characterized is most likely an important and universal feature of many SDR proteins providing a common essential means for substrate binding, processing, and product release. Furthermore, the lid domain displays both sequence variations and structural diversity among the SDR family members, thereby providing a mechanistic basis for differing selectivity among different SDR proteins[1]. While the evolutionary divergence of this domain likely directs substrate selection, selectivity is not absolute, as each SDR protein catalyzes the same dehydrogenase activity on a wide

spectrum of distinct substrates[23]. As an example, the plasticity of the substrate binding lid within 15-PGDH provides a molecular basis for 15-PGDH to enzymatically oxidize multiple related eicosanoid substrates that includes several prostaglandins as well as lipoxin A4 and resolvin E1[24,25].

PGE2 and other eicosanoids are generated through metabolism of arachidonic acid by the cyclooxygenase enzymes prostaglandin-endoperoxide synthase 1 and 2, which generate PGH2, which is a substrate for subsequent synthases, such as the prostaglandin E synthases that generate PGE2[2]. PGE2 degradation is initiated by 15-PGDH catalyzed oxidation of the PGE2 15-hydroxyl group to generate the 15-keto-PGE2 metabolite, a reaction that is rate limiting for PGE2 catabolism[1,2]. Subsequent ß- and ω-oxidation reactions generate 11α-hydroxy-9,15-dioxo-2,3,4,5-tetranor-prostane-1,20-dioic acid (PGE-M), the principal PGE2 urinary metabolite[26]. Studies of the 15-PGDH knockout mouse and of mice treated with (+)-SW209415 or with SW222746 demonstrate that inhibiting 15-PGDH induces accumulation of tissue PGE2 and up-regulation of tissue repair capacity across multiple organs, confirming the central role of this pathway in regulating in vivo physiology[1,3–5,13,27]. This study now illuminates the shared structural mechanism by which this key regulatory enzyme interacts with both small molecule inhibitors and physiologic substrate and provides a structural roadmap for designing the next generation of small molecules for modulating 15-PGDH enzyme function.

In summary, our study depicts a bona fide solution structure of a 15-PGDH-inhibitor complex to define the molecular details of impeding enzyme activity. It provides insights into the structural basis for designing in vivo active small molecule 15-PGDH inhibitors for potential therapeutic use and sheds light on the fundamental dynamics of the enzyme that most likely play key roles in its physiologic function. Finally, our investigation further highlights the rapidly growing role and importance of cryo-EM in the drug discovery pipeline, as the capabilities for high-resolution structure determination are expanded to include those that are at least as small as 29 kDa, albeit in the context of a 58 kDa homodimeric complex.

## Methods

### Expression and purification of human 15-PGDH and mutants
15-PGDH was prepared as previously described[1]. The pET-28b vector was modified to express 15-PGDH with the addition of a C-terminal 6xHis tag with a TEV cleavage site (sequence: GSKENLYFQGHHHHHH) and MH to MAH sequence variation at the N-terminus. The protein was expressed in E. coli BL21(DE3)-Rosetta cells and purified using immobilized Ni$^{2+}$-affinity chromatography. The affinity-purified sample was >95% pure as evaluated by SDS-PAGE gel. Purified recombinant 15-PGDH protein was frozen at 4.5 mg/mL protein in the buffer with 50 mM Tris-HCl pH 7.5, 0.5 mM DTT and 10% glycerol, and stored as aliquots at −80 °C until use. 15-PGDH mutants F185A, Y217A and F185A/Y217A double mutation were generated using the Q5 Site-Directed Mutagenesis Kit (NEB). The variants were expressed and purified identically to the wild-type protein as described above except TALON® Superflow™ (TAKARA) was used for affinity pull down of recombinant proteins.

### Activity assay of drug-bound 15-PGDH protein
15-PGDH activity was assayed as previously described[1]. For $K_m$ determination for PGE2, the reaction mixture contained 50 mM Tris-HCl (pH 7.5), 0.1 μM DTT, 300 μM NAD+, 6 nM of purified enzyme, and PGE2 in a final volume of 200 μL. Six different concentrations of PGE2, ranging from 1.25 μM to 40 μM, and a fixed concentration of NAD+ at 300 μM were used to determine $K_m$ value for PGE2. The reaction mix was measured immediately using an Envision Reader (PerkinElmer). Enzyme activity at each concentration of PGE2 was assayed by determining the slope of generation of NADH as assessed by recording fluorescence at Ex/Em = 340 nM/485 nM at 15 sec increments,

commencing immediately after addition of PGE2, and followed for 105 s (8 reads). Calculations were performed using the SLOPE function in Excel that determines the best fit linear regression line through the data points. Each concentration was assayed in triplicate, and $K_m$ and Hill coefficients were determined using GraphPad Prism 9 software. For IC50 determination, kinetics reactions containing 2 nM 15-PGDH enzyme (wild type and Y217A mutant only), 300 μM of NAD+, 40 μM PGE2, and varying concentrations of drugs ((+)-SW209415 or SW222746) from 0.078 nM to 500 nM were mixed in a total volume of 200 μL in reaction buffer (50 mM Tris-HCl pH 7.5, 0.01% Tween 20, 0.1 μM DTT). Inhibitor was incubated for 5 min with 15-PGDH enzyme before the addition of PGE2. Enzyme activity was determined by following generation of NADH as assayed by recording fluorescence at Ex/Em=340 nM/485 nM for 9 min in 1 min increments, commencing immediately after the addition of PGE2. Percentage of inhibition relative to the reaction in the absence of drugs were plotted against drug concentrations and fitted to the dose-response curve (three parameters) using GraphPad Prism 9 software.

### Protein thermal shift assay
15-PGDH thermal denaturation assays were performed as previously described[1]. Thermal denaturation of 15-PGDH was monitored by differential scanning fluorimetry using SYPRO orange dye[28]. Briefly, the protein was diluted to a final assay concentration of 10 μM in 100 mM Tris buffer pH 7.5, containing 0.01% Tween 20, 0.5 μM DTT and 1:1000 SYPRO orange dye (Sigma S-5692). The final assay volume was 20 μL, with or without 125 μM of NADH. (+)-SW209415 or SW222746, in assay buffer plus 0.4% (v/v) DMSO, was added to 40 μM final concentration. Heat denaturation curves were recorded using a real-time PCR instrument (CFX-96, Bio-Rad) applying a temperature gradient of 2 °C/min. Analysis of the data was performed using default Bio-Rad CFX Manager V3.1 software. Melting temperatures of 15-PGDH were determined by the inflection points of the plots of −d(RFU)/dT.

### Binding of PGDH protein and (+)-SW209415 or SW222746 for cryo-EM grid preparation
Each vial of stocked PGDH was thawed on ice for five minutes. Samples were spun down and the 15-PGDH protein concentration was measured using a bicinchoninic acid (BCA) kit (Thermo Scientific, cat. #23225). 8.4 μM 15-PGDH was mixed with 36 μM (+)-SW209415 and 100 μM NADH in 100 μl for 15 min at room temperature and avoiding light. The mixture was subjected to size-exclusion chromatography (Superdex 75 column; GE Healthcare Life Sciences) with 18 μM (+)-SW209415 and 100 μM NADH present in the running buffer (PBS: pH 7.4, 0.1 mM TCEP). The peak fractions were collected and BCA detecting the 15-PGDH protein concentration was collected for grid preparation and 15-PGDH enzyme activity detection. For SW222746, 15 μM 15-PGDH protein was mixed with 60 μM of SW222746 in 100 μl for 15 min at room temperature avoiding light. The mixture was subjected to size-exclusion chromatography with 30 μM SW222746 present in the running buffer. The peaks were collected and 15-PGDH protein concentration was detected by BCA for grid preparation and 15-PGDH enzyme activity detection.

Cryo-EM grids (Quantifoil AU 1.2/1.3 300 mesh) were glow-discharged and coated with graphene oxide thin layer flakes following the protocol from ref. [29] (figshare. Media. https://doi.org/10.6084/m9.figshare.3178669.v1). Grids were then blotted with purified 15-PGDH with NADH or with NADH and inhibitor, before plunge freezing in liquid ethane. The cryo-EM specimens were prepared using an FEI Vitrobot Mark IV with 3.5 μl of freshly purified protein complex. Grids were blotted for 2.5 s with blot force 0 in 100% humidity at 4 °C prior to plunging freezing. The frozen-dehydrated grids were transferred to a Titan Krios (Thermo Fisher Scientific) transmission electron microscope equipped with a Gatan K3 direct-electron counting camera and BioQuantum energy filter for data acquisition. Movies of the specimen

were recorded with a nominal defocus setting in the range of −1.0 to −1.8 µm using SerialEM with beam-tilt image-shift data collection strategy with a 3 × 3 pattern and 3 shots per hole. The movie stacks were collected in the correlated double sampling (CDS) super-resolution mode of the K3 camera at a nominal magnification of 165,000, yielding a physical pixel size of 0.52 Å/pixel. Each stack was exposed for 4 s, with each frame exposed for 0.1 s, resulting in a 40-frame movie. For datasets without using CDS mode, the movie stacks were collected in the super-resolution mode at a nominal magnification of 165,000 with an exposure time of 2 s and each frame exposed for 0.05 s. The total accumulated dose on the specimen was ~148 electrons per Å² for each stack.

### Image processing of movie stacks

Each movie stack was processed on-the-fly using CryoSPARC live (v2.14.3-beta & v3.2.0)[30,31]. The movie stacks were aligned using patch motion correction with a F-crop factor of 0.5. The contrast-transfer function (CTF) parameters of each particle were estimated using patch CTF. Particles were autopicked using a 100 Å gaussian blob. The numbers of bin2 particles selected after 2D classification are included in Supplementary Table 2. The initial 3D volume and decoys were generated using ab initio reconstruction with a minibatch size of 1000 using selected and unselected 2D classes, respectively. The particles after 2D clean up were submitted to one round of heterogeneous refinement with ab initio 3D volume from good 2D classes and decoy 3D volumes from bad 2D classes. Based on the coordinates and angular information of these particles, bin1 particles of the 3D class with well-resolved secondary structure features were re-extracted from the dose-weighted micrographs. The final particle set was subjected to non-uniform 3D refinements[31], followed by global CTF and local CTF refinements, and local 3D refinements, yielding finals maps with reported global resolutions using the 0.143 criterion of the gold-standard Fourier shell correlation (FSC) (Supplementary Table 2). For the NADH-bound cryo-EM map, the initial 3D volume was generated using ab initio reconstruction with maximum resolution set to 8 Å, and the non-uniform 3D refinement was performed with an initial lowpass resolution of 8 Å to preserve map features reconstituted in the ab initio 3D volume. The half maps were used to determine the local resolution of each map using Relion 3.0[32,33].

### Model building and refinement

The initial model was built ab initio with protein sequence by manually placing individual amino acids into the (+)-SW209415-bound cryo-EM map using Coot[34,35]. Three-dimensional models for small molecule compounds were built and energy minimized in Molecular Operating Environment (MOE)[36], optimized ligand geometry was used as the input for eLBOW[37] to generate restraints for refinement in Phenix or Coot. Density modification maps were generated with the half-maps[38] and used for subsequent real-space refinement in Phenix[39]. The refined structure models were subjected to manual examination and adjustment using Coot[34]. Model overfitting was evaluated against one cryo-EM half map in Phenix[40]. Q-scores for side-chains were calculated using MapQ plugin (https://github.com/gregdp/mapq) in UCSF Chimera[8]. Structural figures were rendered using UCSF Chimera[41]. The final refinement statistics for all three models are provided in Supplementary Table 2.

### Molecular dynamics (MD) simulation setup

MD simulations were performed using the NAMD[42] and the amber ff14sb[43], ions[44] with the TIP3P water model[45]. Parameters for NADH and (+)-SW209415 and S222746 inhibitors were from the Generalized Amber Force Field 2 (GAFF2) and partial charges were calculated using ANTECHAMBER with the AM1-bcc model[46–49]. The starting structures were prepared using the LEaP module in AMBER18 package: proteins were solvated in a cubic water box with a 16 Å padding in all

directions[48]. Two sodium ions were placed in the most negatively charged positions around the protein to neutralize charges of the systems, and 64 sodium ions and 64 chloride ions were added to achieve a physiological salt condition of 150 mM. The systems were energy minimized for 10,000 steps to remove bad contacts. Then, the systems were equilibrated with all heavy atoms restrained harmonically and the temperature raised 10 K per 10,000 steps starting from 0 K to 300 K using temperature reassignment. After reaching the desired temperature, harmonic restrains were gradually reduced using scale from 1.0 to 0 with 0.2 decrement for every 50,000 steps. MD simulations were performed under the NPT ensemble[50,51]. Langevin dynamics was used for constant temperature control, with the value of Langevin coupling coefficient and the Langevin temperature set to 5 ps and 300 K respectively. The pressure was maintained at 1 atm using the Langevin piston method with a period of 100 fs and decay times of 50 fs. A time step of 2 fs was used for all the simulation by using the SHAKE algorithm[52] to constrain bonds involving hydrogen atoms.

Well-equilibrated structures after 100 ns of MD simulation were used as inputs for simulations on Anton 2. Amber topology file and restart file including coordinates and velocities were used to convert into dms input file for Anton 2 using v2software/1.48.0c7. All simulations were also performed following the same MD protocol under the NPT ensemble at 310 K but with a Berendsen thermostat and barostat[53]. Snapshots were saved for every 240 ps. A summary of the simulations performed using Anton 2 is presented in Supplementary Table 3.

### Relative energy contribution with MM/PBSA calculations

For each snapshot, every 1 ns of the 1.5 µs trajectory, of ligand and protein-complex, the binding energy of MM/PBSA was calculated using Eq. (1):[54,55]

$$\triangle G_{binding} = G_{complex} - G_{protein} - G_{ligand}$$
$$= \triangle E_{MM} + \triangle G_{PB} + \triangle G_{nonpolar} - T\triangle S \quad (1)$$

where $\Delta E_{MM}$ is the molecular mechanic (MM) interaction energy calculated in gas-phase between protein and ligand, including electrostatic and van der Waals energies; the desolvation-free energy consists of polar ($\Delta G_{PB}$) and nonpolar ($\Delta G_{nonpolar}$) terms; $T\Delta S$ is the change of conformational entropy on ligand binding, which was not considered here as the ligand binding pocket is very stable and the comparison was performed internally. The decomposition of the binding free energy to the relative energy contribution from individual residues was performed using the MM-PBSA.py module in AMBER18[56].

### Hydroxyl radical footprinting and mass spectrometry analysis

Synchrotron hydroxyl radical footprinting experiments were performed at beamline X28C of the National Synchrotron Light Source at Brookhaven National Laboratory. Samples containing 5 µM of 15-PGDH protein (+)-SW209415 were exposed for 0–75 milliseconds at ambient temperature and immediately quenched with methionine amide at the 10 mM final concentration to prevent secondary oxidation[57]. All protein samples were then reduced with 10 mM dithiothreitol (DTT) at 56 °C for 45 min and alkylated with 25 mM iodoacetamide at room temperature and for 45 min. Protein samples then were digested with trypsin (Promega, Madison, WI) at 37 °C for overnight followed by Asp-N digestion at 37 °C for 8 h. with an enzyme:protein molar ratio of 1:10. The digestion reaction was terminated by heating samples at 95 °C for 2 min.

Identification and quantification of oxidative sites were performed by liquid chromatography-mass spectrometry (LC-MS) analysis using an Orbitrap Elite mass spectrometer (Thermo Electron, San Jose, CA) interfaced with a Waters nanoAcquity UPLC system (Waters, Taunton, MA). A total of 250 ng of proteolytic peptides were loaded on a trap column (180 µm × 20 mm packed with C18 Symmetry, 5 µm,

100 Å (Waters, Taunton, MA)) to desalt and concentrate peptides, and subsequently eluted on a reverse phase column (75 µm × 250 mm nano column, packed with C18 BEH130, 1.7 µm, 130 Å (Waters, Taunton, MA)) using a gradient of 2 to 42% mobile phase B (100% acetonitrile/ 0.1% formic acid) vs. mobile phase A (100% water/0.1 % formic acid) over a period of 60 min at 37 °C with a flow rate of 300 nl/min. Peptides eluting from the column were introduced into the nano-electrospray source at a capillary voltage of 2.4 kV. All MS data were acquired in the positive ion mode. For MS1 analysis, a full scan was recorded for eluted peptides (m/z range of 360–1600) in the Orbitrap mass analyzer with resolution of 120,000 followed by MS/MS of the 20 most intense peptide ions scanned in the ion trap mass analyzer. Selected ion currents for modified and unmodified peptides in MS1 experiments were used to determine the extent of oxidation for each modified site. The resulting MS/MS data were searched against human 15-PGDH database using Mass Matrix and Origin 8.0 software to identify sites of modification and to quantify modification rates, respectively. In particular, MS1 and MS/MS spectra were searched for peptides generated from 15-PGDH protein sequence by dual trypsin and Asp-N digestions using mass accuracy values of 10 ppm and 0.7 Daltons for MS1 and MS/MS scans, respectively, with allowed variable modifications including carbamidomethylation for cysteines and all known oxidative modifications previously documented for amino acid side chains. In addition, MS/MS spectra for each site of proposed modification were manually examined and verified.

### Calculation of modification rate and protection factor for a specific site

The integrated peak areas of the unmodified peptide ($A_u$), and of a peptide in which a residue is modified ($A_m$) derived from selected ion chromatograms, were used to calculate the fraction unmodified for each specific modified species according to Eq. (2):

$$F_u = 1 - (A_m/(A_u + \sum A_m)) \qquad (2)$$

where $\sum A_m$ is the sum of all modified products for a particular peptide. Dose–response curves were generated using fraction unmodified values for each specific site of modification plotted versus X-ray exposure time. The fraction unmodified for each site of modification was fit to Eq. (3):

$$F_u(t) = F_u(0)e^{-kt} \qquad (3)$$

where $F_u(0)$ and $F_u(t)$ are the fraction of unmodified at time 0 and time $t$, respectively, and $k$ is a first-order modification rate constant. The modification rate constants of residues for which rates were determined are provided in Supplementary Table 1. The Protection Factor (PF) of individual residues with modification rates was calculated using Eq. (4)[11]:

$$PF_i = R_i/k_i^{fp} \qquad (4)$$

where $R_i$ is the intrinsic reactivity of residue $i$ to hydroxyl radicals[58], and $k_{fp}$ is the modification rate for residue $i$ as shown in Supplementary Table 1. Rate constants were calculated based on duplicate reactions that were each sampled at four different time points, providing eight independent data elements[59].

### Reporting summary

Further information on research design is available in the Nature Portfolio Reporting Summary linked to this article.

## Data availability

The cryo-EM maps of (+)-SW209415 and SW222746-bound to 15-PGDH with NADH and the apo-form with NADH have been deposited into the Electron Microscopy Data Bank under accession codes EMD-27010, EMD-27025 and EMD-29005, respectively. The corresponding atomic models have been deposited into Protein Data Bank under accession codes 8CVN, 8CWL and 8FD8. Cryo-EM grid preparation is detailed here (https://doi.org/10.6084/m9.figshare.3178669.v1). Source data are provided with this paper.

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

## Acknowledgements

We thank Dr. Hsin-Hsiung (Daniel) Tai for seminal contributions to understanding 15-PGDH and for help in establishing 15-PGDH enzyme kinetic assays. Computational support was provided by the Case Western Reserve University High Performance Computing Cluster. Anton 2 computer time was provided by the Pittsburgh Supercomputing Center (PSC) through Grant R01 GM116961 from the National Institutes of Health. The Anton 2 machine at PSC was generously made available by D.E. Shaw Research. Cryo-EM data were collected at, and sponsored by, the National Cancer Institute's National Cryo-EM Facility at the Frederick National Laboratory for Cancer Research under contract HSSN261200800001E, the Midwest Consortium for High-Resolution Cryo-Electron Microscopy (U24 GM116789) and Cryo-EM pilot funds from Case Western Reserve University. This work was primarily supported by a grant from the NIH (RM1 GM142002 to D.J.T., S.D.M., & J.M.R.) and further supported by other NIH grants (R01 GM133841 and R01 CA240993 to D.J.T.; R35 CA197442 and P50 CA150964 to S.D.M.; R01 CA216863 to J.M.R. and S.D.M.; and U54 CA231649 to J.M.R.) and the Welch Foundation (I-1612 to J.M.R.).

## Author contributions

H.L., W.H. and S.R. expressed and purified wild type and mutant recombinant 15-PGDH protein. H.L. and S.P.F. conducted biochemical experiments and analyzed the results. W.H. prepared cryo-EM grids, collected cryo-EM data, solved cryo-EM structures, and built models into the cryo-EM maps. W.H., J.M.R., S.D.M. and D.J.T. analyzed and interpreted the structural and biochemical data. A.D. and W.H. performed molecular dynamic simulations. J.K., Y.Y. and M.C. conducted hydroxyl radical footprinting and mass spectrometry experiments and analyzed the results from those experiments. W.H., J.M.R., S.D.M. and D.J.T. designed and supervised the experiments and wrote the manuscript with input from all authors.

## Competing interests

JMR, SDM, and YY, are inventors on patents describing inhibitors of 15-PGDH and their therapeutic uses. These include awarded US patents: 9790233, 9789116, 9801863, 10301320, 10420752, 10869871, 10945998, 11426420. These patents have been licensed to Rodeo Therapeutics, which is owned by Amgen. JMR and SDM are beneficiaries of founders stock in Rodeo, and serve as consultants to Amgen on topics related to 15-PGDH inhibition. All other authors declare no competing interests.
