## [Peer Review File · Nature Communications]

Small Molecule Inhibitors of 15-PGDH Exploit a Physiologic Induced-Fit Closing SystemREVIEWER COMMENTS

Reviewer #1 (Remarks to the Author):

Huang et. al. report two high resolution structures of 15-PGDH enzyme individually bound with two inhibitors representing distinct chemical scaffold. These two structures are important in two aspects: 1) they reveal an induced-fit mechanism whereby a disordered lid closes around both inhibitors; 2) the structures also provide insights into potential roles of this newly discovered dynamic lid structure in regulating substrate sampling and product release. On the technical front, this manuscript provides further impetus to use cryo-EM to advance drug discovery for proteins as small as 60 kD.

I only have minor comments:

- 1) Figure 3 panel b could be made more accessible to general readers.
- 2) In supplementary Figure 1, a panel showing angular distribution could be included.
- 3) In supplementary Figure 2, density for NADH could be included.
- 4) If the labs have enzymatic assay in place, I would encourage to test several structure inspired mutants that have not been explored before. For example, the dimer interface etc.

Reviewer #2 (Remarks to the Author):

The authors report two cryo-EM structures of small molecule-bound 15-prostaglandin dehydrogenase (15-PGDH), a key negative regulator of tissue stem cell activity in proliferation and repair of tissue injury. As such it is a potential therapeutic target for regenerative medicine, corroborated by small molecule 15-PGDH inhibitors leading to accelerated tissue recovery and repair across animal models. Structures of apo 15-PGDH are known but no inhibitor-bound structures have been deposited into the PDB yet. The authors employed cryo-EM to obtain high-resolution structures of native 15-PGDH and two structures of inhibitor-bound 15-PGDH. Analysis of the structures, in combination with hydroxyl radical footprinting data and MD simulations, enabled identification of ligand binding and ligand gating residues. The authors conclude by speculating that their inhibitor-bound structures will facilitate future drug discovery efforts.

This represents a solid structural biology study, elucidating small molecule-bound 15-PGDH conformations. However, I do not think that the novelty and impact warrant publication in Nature Communications. More specifically, a high-resolution structure of 15-PGDH exists and the manuscript only provides slightly different inhibitor-bound conformations. Additionally, it is not clear that the new inhibitor-bound conformations indeed improve structure-based drug discovery. This manuscript would be more suited for a targeted structural biology journal. Alternatively, and this would require significant additional work through structure-based drug discovery, the authors could opt to show that their structures are indeed providing better drug candidates than for example 2GDZ, the crystal structure of 15-PGDH, complexed with only NAD⁺. Addition of this data would significantly increase the impact of the work.

Minor comment to address in future versions: why did the authors choose to run their MD simulations in NAMD with the amber force field? This seems somewhat uncommon.

Reviewer #3 (Remarks to the Author):

Prostaglandin E2 (PGE2) is a pleiotropic lipid mediator derived through the cyclooxygenase (COX) pathway of arachidonic acid metabolism. Non-steroidal anti-inflammatory drugs target COX and block the synthesis of PGE2 affording analgesic, antipyretic and anti-inflammatory effects. PGE2 is metabolized in two sequential steps by 15-hydroxyprostaglandin dehydrogenase (15-PGDH) followed by 15-

oxoprostaglandin 13-reductase/leukotriene B4 12-hydroxydehydrogenase (PGR/LTB4DH). Inhibition of 15-PGDH has been suggested as a way to maintain PGE2 levels with potential beneficial effects in certain pathological conditions.

Huang et al. presents cryoEM structures of 15-PGDH in complex with its cofactor NADH and two synthetic inhibitors, (+)-SW209415 and SW222746. Although the x-ray structure of 15-PGDH in complex with NAD⁺ has been previously reported, the present study uncovers several novel and interesting structural features of the enzyme, most notably a dynamic lid composed of a triple-helix motif. In its closed conformation, the lid encapsulates the inhibitors and promotes firm binding to the protein. It seems likely that the lid is also involved in enzyme catalysis.

1. The structural data and results of HRF-MS point to two specific residues of the dynamic lid, Phe185 and Tyr217, which seem critical for interactions with the inhibitors and possibly the enzyme substrate, PGE2. The functional importance of these two residues is not well supported by experimental data and should be further validated by site-directed mutagenesis and kinetic analysis.

2. The long MD simulations reveal intriguing differences in dynamic behavior between the lids of the respective enzyme monomer such that one active site may always be open for binding of inhibitor or substrate. Here, the authors could easily strengthen the results by kinetic determination of the stoichiometry between substrate and enzyme protomer.

3. In Extended Data Fig. 10, the authors show a high degree of conservation of Phe185 and Tyr217 among 15-PGDH from multiple species. Are these two residues found among other human short-chain dehydrogenases?

4. The authors should add some text to put their work in the context of arachidonic acid metabolism.

Reviewer #4 (Remarks to the Author):

The manuscript reports a comprehensive study that reveals the structure of 15-PGDH with the presence of small molecule inhibitors. Structural model obtained from Cryo-EM was complemented nicely by the MD simulation and hydroxyl radical footprinting. Overall, the topic is of broad interest and fits in the scope of Nature Communications. I would like to share the following minor comments with the authors regarding on the mass spectrometry-related contents in the manuscript.

1. The manuscript reports another nice example of how MS-based structural tools can be used to address biological questions. HRF reveals the key binding residues between 15-PGDH and (+)-SW209415. As the authors also studied the interaction between 15-PGDH and SW222746, why HRF is not implemented in analyzing this binding pair?

2. In Figure 3b, authors should specify the flipped angle between the two views.

3. Are the HRF experiments performed with replicates? Are the errors reported in the extended data table 1 from the fitting? Or is it a propagated error taken into account the errors of fraction unmodified from any experimental replicates?

4. In Page 22, line 547, authors report an enzyme:protein molar ratio of 1:10. Just want to double check, is it molar ratio or weight ratio?

5. In page 23, authors report the calculation of protection factor and the corresponding results are reported in Extended Data Table 1. Some of the residues have negative protection factors, indicating that those residues show higher reactivity as compared with their intrinsic reactivities. Can authors comment on the reason why the reactivity of some the solvent exposed residues, e.g. Y256, is much higher as compared with their intrinsic reactivities? Is it related to local environment of the residue? Or is it because the intrinsic reactivity is measured based on the reaction between hydroxyl radical and free amino acid rather than amino acid residues?

- 6. In Extended Data Table 1, there are two "C182" entries with different values. Please double check and see if either one of them should be removed.**
- 7. Also in Extended Data Table 1, the authors did not highlight the residues with significant differences as mentioned in the table caption.**

We thank the reviewers for taking the time to evaluate our data and for providing constructive feedback to improve and strengthen the work we presented in our original submission. We have fully revised our manuscript to include the recommended additional new experimental data and to address all of the reviewer' concerns. The revisions and new data are now included in the revised manuscript and are summarized in the point-by-point response below. We highlight new experimental data with site specific mutants of the Y217 and F185 residues, that our cryo-EM data nominated as key residues in the 15-PGDH lid structure. Functional characterization of these mutants now provides further direct demonstration that these two residues play a key functional role in the binding of 15-PGDH to the small molecule inhibitors, and further demonstrate these residues also play a key role in 15-PGDH interaction with its physiologic substrate PGE2.

Reviewer #1 (Remarks to the Author):

Huang et. al. report two high resolution structures of 15-PDGH enzyme individually bound with two inhibitors representing distinct chemical scaffold. These two structures are important in two aspects: 1) they reveal an induced-fit mechanism whereby a disordered lid closes around both inhibitors; 2) the structures also provide insights into potential roles of this newly discovered dynamic lid structure in regulating substrate sampling and product release. On the technical front, this manuscript provides further impetus to use cryo-EM to advance drug discovery for proteins as small as 60 kD.

I only have minor comments:

1) Figure 3 panel b could be made more accessible to general readers.

As suggested by the review, we adjusted Fig. 3b with the protein shown as a blob. Only dimeric interface and the lid are shown in ribbon. Residues observed in HRF experiments are shown as sticks. Most of the protected residues are directly surrounding the drug and NADH nicotinamide moiety, except Y116 and L117, whose protection is due to indirect stabilization of the dimer interface upon lid closing.

2) In supplementary Figure 1, a panel showing angular distribution could be included.

A panel showing the angular distribution of particle set from the final refinement step is now included as ED. Fig1e.

3) In supplementary Figure 2, density for NADH could be included.

The density for NADH was included in ED Fig. 2 of our original submission. However, we've modified the figure to improve contrast/visibility for the mesh density surrounding bound NADH and inhibitor.

4) If the labs have enzymatic assay in place, I would encourage to test several structure inspired mutants that have not been explored before. For example, the dimer interface etc.

In our revised manuscript, we have added a significant amount of functional data to both validate and complement the structural investigation. Namely, as suggested by the reviewer, we introduce alanine substitutions at F185 and Y217, which we had shown to stack with the bound inhibitor to promote and stabilize 'closing' of the 15-PGDH lid. We have characterized the single and double-mutants using biophysical and enzymatic assays to confirm the key role of these conserved residues in 15-PGDH binding of inhibitors, as assayed by changes in the thermal melting point of the inhibitor enzyme complex and as assayed by alterations of inhibitor IC50. Specifically, we have demonstrated that, as predicted, both the F185A and Y217A mutations significantly reduce the thermal stabilization of the inhibitor bound 15-PGDH complex. We have also shown, as predicted, that the Y217A mutant significantly increases the IC50 concentration for the small molecule inhibitors. We have also employed enzymatic assays to demonstrate a key functional role of these same two residues in interaction of 15-PGDH with the native PGE2 substrate, as reflected by 5-fold increased Km of the Y217A mutant and by complete loss of

enzyme activity of the F185 mutant (making it impossible to determine effects of the F185A substitution on IC50). These new data are presented in **Figure 5 and Extended Data Figures** of the revised manuscript.

Unfortunately, a similar mutagenesis approach in probing the dimer interface is complicated due to residues of the closed lid also interacting with residues at the dimer interface on the opposite protomer (e.g. Y116, A168, L171, and M172). Given the dual functions of these residues in both stabilizing the dimer interface and also interacting with the lid, a rigorous separation-of-function of these two events would need to extend well beyond the focus of our current investigation. We will aim to conduct these studies as part of a separate, more focused investigation.

Reviewer #2 (Remarks to the Author):

The authors report two cryo-EM structures of small molecule-bound 15-prostaglandin dehydrogenase (15-PGDH), a key negative regulator of tissue stem cell activity in proliferation and repair of tissue injury. As such it is a potential therapeutic target for regenerative medicine, corroborated by small molecule 15-PGDH inhibitors leading to accelerated tissue recovery and repair across animal models. Structures of apo 15-PGDH are known but no inhibitor-bound structures have been deposited into the PDB yet. The authors employed cryo-EM to obtain high-resolution structures of native 15-PGDH and two structures of inhibitor-bound 15-PGDH. Analysis of the structures, in combination with hydroxyl radical footprinting data and MD simulations, enabled identification of ligand binding and ligand gating residues. The authors conclude by speculating that their inhibitor-bound structures will facilitate future drug discovery efforts.

This represents a solid structural biology study, elucidating small molecule-bound 15-PGDH conformations. However, I do not think that the novelty and impact warrant publication in Nature Communications. More specifically, a high-resolution structure of 15-PGDH exists and the manuscript only provides slightly different inhibitor-bound conformations. Additionally, it is not clear that the new inhibitor-bound conformations indeed improve structure-based drug discovery. This manuscript would be more suited for a targeted structural biology journal. Alternatively, and this would require significant additional work through structure-based drug discovery, the authors could opt to show that their structures are indeed providing better drug candidates than for example 2GDZ, the crystal structure of 15-PGDH, complexed with only NAD⁺. Addition of this data would significantly increase the impact of the work.

While it is true, as this reviewer states, that a high-resolution structure of 15-PGDH exists, our investigation extends far beyond what is known from that prior study. In the x-ray structure of 15-PGDH, the engineered TEV cleavage site of 15-PGDH is trapped in the drug/substrate binding pocket with the lid of the enzyme ordered and closed around it. This finding leaves the false impression that the lid of 15-PGDH is presumably always ordered and closed and, therefore, does not allow for functional interpretation that is important for drug design and for understanding the mechanism of 15-PGDH enzymology. Our study reconciles this notion by showing that the lid of 15-PGDH is indeed dynamic in the absence of inhibitor/substrate and this dynamic nature is critical for enzyme function. As such, the previous x-ray structure does not serve as a viable tool for structure-based drug design, which has prevented the structure of ANY inhibitor being bound to the enzyme. Our manuscript now includes the structure of two, chemically unrelated inhibitors bound to the enzyme, with the key interactions needed for binding elucidated. Moreover, the key finding of our paper extends well beyond the structure of an enzyme with the primary finding focusing on the molecular mechanism of lid closing and its relationship to inhibitor binding that we have identified. Additionally, our manuscript defines the key role of lid residues F185 and Y217, and of the novel lid mechanism, in 15-PGDH interaction both with inhibitors and with the native PGE2 substrate. We have attempted to better articulate all of these points in the revised manuscript.

Minor comment to address in future versions: why did the authors choose to run their MD simulations in NAMD with the amber force field? This seems somewhat uncommon.

We respectfully disagree with the reviewer on this topic (or we misunderstand the reviewer's intended statement). NAMD is a commonly used MD simulation integrator that is well known for its linear scalability. Accordingly, using NAMD allows us to obtain the same length of simulation time with much less computational wall time. Since the early days of its invention, NAMD has been used with AMBER force field (see, for example, PMID: 16170796) and the instructions for doing so are well documented in the NAMD manual:

<https://www.ks.uiuc.edu/Research/namd/2.14/ug/node13.html>

We also refer to Dr. David Case's excellent summary that focuses on MD simulations and highlights such a combination that we have used:

https://ambermd.org/namd/namd_amber.html

Reviewer #3 (Remarks to the Author):

Prostaglandin E2 (PGE2) is a pleiotropic lipid mediator derived through the cyclooxygenase (COX) pathway of arachidonic acid metabolism. Non-steroidal anti-inflammatory drugs target COX and block the synthesis of PGE2 affording analgesic, antipyretic and anti-inflammatory effects. PGE2 is metabolized in two sequential steps by 15-hydroxyprostaglandin dehydrogenase (15-PGDH) followed by 15-oxoprostaglandin 13-reductase/leukotriene B4 12-hydroxydehydrogenase (PGR/LTB4DH). Inhibition of 15-PGDH has been suggested as a way to maintain PGE2 levels with potential beneficial effects in certain pathological conditions.

Huang et al. presents cryoEM structures of 15-PGDH in complex with its cofactor NADH and two synthetic inhibitors, (+)-SW209415 and SW222746. Although the x-ray structure of 15-PGDH in complex with NAD⁺ has been previously reported, the present study uncovers several novel and interesting structural features of the enzyme, most notably a dynamic lid composed of a triple-helix motif. In its closed conformation, the lid encapsulates the inhibitors and promotes firm binding to the protein. It seems likely that the lid is also involved in enzyme catalysis.

Indeed, as suggested by the reviewer, our added functional data indicate that an analogous role of 15-PGDH lid closing is important not only for binding inhibitors but equally for enzyme binding to and activity against native substrates. We have described the new data throughout the revised manuscript to emphasize this point and thank the reviewers for this insight.

1. The structural data and results of HRF-MS point to two specific residues of the dynamic lid, Phe185 and Tyr217, which seem critical for interactions with the inhibitors and possibly the enzyme substrate, PGE2. The functional importance of these two residues is not well supported by experimental data and should be further validated by site-directed mutagenesis and kinetic analysis.

This same suggestion was provided by reviewer 1. In response we have now added the suggested new experiments involving site specific mutation of these residues and the functional characterization of their effect on interaction with inhibitors (as assessed by thermal melting and by IC50), and on interaction with PGE2 (as assessed by enzyme kinetics). These new studies are now presented in the revised manuscript and detailed above in our response to Reviewer 1, statement 4.

2. The long MD simulations reveal intriguing differences in dynamic behavior between the lids of the respective enzyme monomer such that one active site may always be open for binding of inhibitor or substrate. Here, the authors could easily strengthen the results by kinetic determination of the stoichiometry between substrate and enzyme protomer.

The reviewer brings up the important consideration that long time scale molecular dynamic (MD) simulations suggested that the lids on the opposite protomers might differ in the poses adopted during fluctuations. However, our enzyme kinetics studies showed the relation of enzyme activity versus PGE2 concentration fits a simple

sigmoidal curve characterized by Hill coefficient of 1. It is thus most likely that the two protomers simply act independently in their binding to inhibitor or to substrate. We have added this information to the discussion.

3. In Extended Data Fig. 10, the authors show a high degree of conservation of Phe185 and Tyr217 among 15-PGDH from multiple species. Are these two residues found among other human short-chain dehydrogenases?

While these two residues are not precisely conserved across SDRs, there are analogous residues that would be expected to fulfil similar roles in regulating lid closures for those other proteins.

4. The authors should add some text to put their work in the context of arachidonic acid metabolism.

Further text regarding arachidonic acid metabolism has been added to the discussion (Page 18-19).

Reviewer #4 (Remarks to the Author):

The manuscript reports a comprehensive study that reveals the structure of 15-PGDH with the presence of small molecule inhibitors. Structural model obtained from Cryo-EM was complemented nicely by the MD simulation and hydroxyl radical footprinting. Overall, the topic is of broad interest and fits in the scope of Nature Communications. I would like to share the following minor comments with the authors regarding on the mass spectrometry-related contents in the manuscript.

1. The manuscript reports another nice example of how MS-based structural tools can be used to address biological questions. HRF reveals the key binding residues between 15-PGDH and (+)-SW209415. As the authors also studied the interaction between 15-PGDH and SW222746, why HRF is not implemented in analyzing this binding pair?

Unfortunately, these experiments are laborious, time-consuming, expensive, and require machine intensive techniques. Additionally, the specialized equipment is available at only a few synchrotron sources, which limits accessibility of conducting these experiments. In designing our experiments, we started with the structure of SW209415 bound to 15-PGDH and proceeded to use HRF to validate our cryo-EM data, while simultaneously pursuing the structure of SW222746 bound to 15-PGDH. Since both results supported our initial structural studies, we felt that the orthogonal approach, particularly combined with our added functional assays, validated our findings and we deemed it not cost-effective to conduct HRF experiments for both inhibitors.

2. In Figure 3b, authors should specify the flipped angle between the two views.

Thank you for this suggestion. We have modified the figure to include the flipped angle of different views to be 90°. The figure has been further revised to incorporate suggestions from another reviewer (R1). Please see our response above to Reviewer 1, Statement 1.

3. Are the HRF experiments performed with replicates? Are the errors reported in the extended data table 1 from the fitting? Or is it a propagated error taken into account the errors of fraction unmodified from any experimental replicates?

The HRF-MS experiments were performed in duplicate. For each residue the modification constants are calculated from determinations at 4 time points that are all assessed in each of the duplicates, or 8 time points total. The errors reported in the Extended Data Table 1 were generated from the fitting of the dose-response plots to the first order kinetic equation. The dose-response plot for each specific residue was generated by averaging duplicate experiments. This approach is the standard in the field (as per: Kiselar J, Chance MR. High-Resolution Hydroxyl Radical Protein Footprinting: Biophysics Tool for Drug Discovery. *Annu Rev Biophys.* **47**, 315-33 (2018)).

We have added this additional information to the description of the methods.

4. In Page 22, line 547, authors report an enzyme:protein molar ratio of 1:10. Just want to double check, is it molar ratio or weight ratio?

We thank reviewer to ask for clarification. It is molar ratio. We've updated the text in the revised manuscript to include this information.

5. In page 23, authors report the calculation of protection factor and the corresponding results are reported in Extended Data Table 1. Some of the residues have negative protection factors, indicating that those residues show higher reactivity as compared with their intrinsic reactivities. Can authors comment on the reason why the reactivity of some the solvent exposed residues, e.g. Y256, is much higher as compared with their intrinsic reactivities? Is it related to local environment of the residue? Or is it because the intrinsic reactivity is measured based on the reaction between hydroxyl radical and free amino acid rather than amino acid residues?

Negative protection factors are well reported in the literature and were observed across a set of residues, all of which map to the protein surface, and should reflect effects of the local environment at the protein - water interface that increase either effective solvent accessibility or residue reactivity. Importantly, negative protection factors do not interfere with the main objective of the study, which was to measure the shift in protection factor upon drug binding.

6. In Extended Data Table 1, there are two "C182" entries with different values. Please double check and see if either one of them should be removed.

We thank the reviewer for bringing up this important point. Two different protection values reported in the Extended Table 1 were calculated for C182 residues labeled with one and two oxygens. Since C182 exhibited the predominant labeling with one oxygen (+16), we removed the protection value corresponding to the labeling of C182 with two oxygens (+32). The +32 modification is listed first in the Table that has now been revised to clarify this point.

7. Also in Extended Data Table 1, the authors did not highlight the residues with significant differences as mentioned in the table caption.

We have corrected this in the revised manuscript.

REVIEWERS' COMMENTS

Reviewer #1 (Remarks to the Author):

The authors have addressed my minor concerns. Overall, this is a solid study combining structural biology, enzymatic assays, and dynamic measurements to provide insights into pharmacology and enzymology of an important therapeutic target in regenerative medicine.

Reviewer #2 (Remarks to the Author):

The authors did not address my main comment asking to use their structure for drug discovery purposes and thus validating their major claim that their inhibitor-bound structures will facilitate future drug discovery efforts. As such, I am still thinking that this work is better suited for a more specialized journal.

Reviewer #3 (Remarks to the Author):

The authors have made substantive additional experiments, most notably site-directed mutagenesis of the key residues implicated in lid regulation of 15-PGDH. Together with textual revisions, they have significantly improved the quality of their work. All my points have been satisfactorily addressed.

Reviewer #4 (Remarks to the Author):

The authors have addressed all my comments and I believe the manuscript is now in good shape. The only minor comment that I have is about the Extended Data Table 1. The authors are highlighting the residues with significant difference, and between the two C182 lines, seems to me that the upper one ($\ln PF = 0.829$) should be highlighted, rather than the lower one ($\ln PF = -0.115$). Thanks!

We again thank the reviewers and the editors for their valuable feedback on our revised manuscript. We have made some final changes to address any lingering concerns, which are included in the point-by-point response to all four reviewers below.

REVIEWERS' COMMENTS

Reviewer #1 (Remarks to the Author):

The authors have addressed my minor concerns. Overall, this is a solid study combining structural biology, enzymatic assays, and dynamic measurements to provide insights into pharmacology and enzymology of an important therapeutic target in regenerative medicine.

Thank you, we appreciate this feedback.

Reviewer #2 (Remarks to the Author):

The authors did not address my main comment asking to use their structure for drug discovery purposes and thus validating their major claim that their inhibitor-bound structures will facilitate future drug discovery efforts. As such, I am still thinking that this work is better suited for a more specialized journal.

We respect this reviewer's opinion and thank them for endorsing the quality of the work presented in our study. Without question, the structures of protein-inhibitor complexes have been, and will continue to be, incredibly valuable for the design and development of rationally designed new inhibitor classes and derivatives. Our study provides this information for two chemically distinct scaffolds bound to 15-PGDH at some of the highest resolutions ever reported for an enzyme/inhibitor cryoEM structure. We also determine the calculated contributions of individual protein side-chains for binding each inhibitor, which could not have been done without the structures. While it is clearly beyond the scope of the present study, we definitely plan in the coming years to continue the work of building off our structures to design the next generation of inhibitors, and certainly hope to be able to report those results in a future publication.

Reviewer #3 (Remarks to the Author):

The authors have made substantive additional experiments, most notably site-directed mutagenesis of the key residues implicated in lid regulation of 15-PGDH. Together with textual revisions, they have significantly improved the quality of their work. All my points have been satisfactorily addressed.

Thank you, we appreciate this feedback.

Reviewer #4 (Remarks to the Author):

The authors have addressed all my comments and I believe the manuscript is now in good shape. The only minor comment that I have is about the Extended Data Table 1. The authors are highlighting the residues with significant difference, and between the two C182 lines, seems to me that the upper one ($\Delta\ln PF = 0.829$) should be highlighted, rather than the lower one ($\Delta\ln PF = -0.115$).

We are grateful that this reviewer identified this minor error. We have made the appropriate correction.